# SCC3 is an axial element essential for homologous chromosome pairing and synapsis

**Yangzi Zhao**[1,2†]**, Lijun Ren**[3†]**, Tingting Zhao**[3†]**, Hanli You**[1]**, Yongjie Miao**[1]**, Huixin Liu**[2]**, Lei Cao**[2]**, Bingxin Wang**[2]**, Yi Shen**[2]**, Yafei Li**[2]**, Ding Tang**[2]**, Zhukuan Cheng**[1]*****

[1]Jiangsu Key Laboratory of Crop Genomics and Molecular Breeding/Key Laboratory of Plant Functional Genomics of the Ministry of Education, Jiangsu Co-Innovation Center for Modern Production Technology of Grain Crops, Yangzhou University, Yangzhou, China; [2]State Key Lab of Plant Genomics, Institute of Genetics and Developmental Biology, Innovation Academy for Seed Design, Chinese Academy of Sciences, Beijing, China; [3]College of Horticulture Science and Engineering, Shandong Agricultural University, Shandong, China

**\*For correspondence:**
zkcheng@genetics.ac.cn

[†]These authors contributed equally to this work

**Competing interest:** The authors declare that no competing interests exist.

## Abstract
Cohesin is a multi-subunit protein that plays a pivotal role in holding sister chromatids together during cell division. Sister chromatid cohesion 3 (SCC3), constituents of cohesin complex, is highly conserved from yeast to mammals. Since the deletion of individual cohesin subunit always causes lethality, it is difficult to dissect its biological function in both mitosis and meiosis. Here, we obtained *scc3* weak mutants using CRISPR-Cas9 system to explore its function during rice mitosis and meiosis. The *scc3* weak mutants displayed obvious vegetative defects and complete sterility, underscoring the essential roles of SCC3 in both mitosis and meiosis. SCC3 is localized on chromatin from interphase to prometaphase in mitosis. However, in meiosis, SCC3 acts as an axial element during early prophase I and subsequently situates onto centromeric regions following the disassembly of the synaptonemal complex. The loading of SCC3 onto meiotic chromosomes depends on REC8. *scc3* shows severe defects in homologous pairing and synapsis. Consequently, SCC3 functions as an axial element that is essential for maintaining homologous chromosome pairing and synapsis during meiosis.

## eLife assessment

This **fundamental** study elucidates the function of the cohesin subunit SCC3 in maintaining homologous chromosome pairing and synapsis during meiosis. The observation of sterility in the SCC3 weak mutant prompted an investigation of abnormal chromosome behavior during anaphase I, and the discovery that SCC3's loading onto meiotic chromosomes is REC8-dependent. The **convincing** evidence presented in this study contributes to our understanding of meiosis in rice and attracts cell biologists, reproductive biologists, and plant geneticists.

## Introduction

The correct segregation of sister chromatids is essential for mitotic cell stabilization and to generate haploid gametes during meiosis. The cohesin complex is a conserved multi-subunit protein system that guarantees the proper segregation of sister chromatids in mitosis across living organisms. This complex is formed by four core subunits, including the structural maintenance of chromosomes (SMC) subunits SMC1 and SMC3, and the sister chromatid cohesin (SCC) subunits SCC1 and SCC3

(*Bolaños-Villegas et al., 2017*; *Moronta-Gines et al., 2019*). The cohesin complex forms a ring encircling the sister chromatids and ensures genomic stability during DNA replication (*Higashi et al., 2020*). In addition, cohesin also participates in the repair of DNA double-strand breaks (DSB), homologous pairing, construction of the synaptonemal complex (SC), orientation of kinetochores, and regulation of gene expression (*Nasmyth and Haering, 2009*).

Cohesin is established during the S phase and includes arm and centromeric cohesin. Arm cohesin ensures sister chromatids bond together during mitotic prophase. Centromeric cohesin persists until the metaphase-to-anaphase transition, and is vital to resist the force exerted by spindle microtubules, allowing for the accurate segregation of sister chromatids to opposite poles of the cell. The dissociation of cohesin generally occurs in two steps, including through the dissociation of cohesin on chromosome arms during prometaphase, followed by the degradation of cohesin at the centromeres at the onset of anaphase (*Ishiguro and Watanabe, 2007*; *Watanabe, 2005*). This allows sister chromatids to segregate in mitosis.

Meiosis contains a single round of DNA replication, but, in contrast to mitosis, is followed by two successive cell divisions. Nonetheless, while the loading mechanism of cohesin is very similar to the analogous process during mitosis, the cohesin degradation during meiosis occurs in two steps: from the chromosome arms during the first meiotic division, and from the centromeres during the second meiotic division (*Mercier et al., 2015*). During meiosis I, homologous chromosomes are linked by chiasmata that separate at metaphase I. At this stage, cohesin dissociates from chromosome arms while the two sister chromatids remain closely together because centromeric cohesin is preserved. Subsequently, the sister chromatids separate at anaphase II due to the complete cohesin disassociation at both chromosome arms and centromeres. During this process, sister chromatid cohesin is protected by shugoshin from cleavage by separases (*Wang et al., 2012*; *Wang et al., 2011*), which ensures the correct separation of sister chromatids during meiosis.

Notably, the cohesin complex plays additional functional roles in meiosis. Specifically, cohesin on chromosome arms is crucial for normal progression of meiosis I, while centromeric cohesin plays a more prominent role in meiosis II by geometrically orientating the sister chromatids. During meiosis I, homologous chromosomes are segregated in opposite directions to halve the chromosome number, but sister chromatids remain temporarily mono-oriented and move to the same pole. In meiosis II, sister chromatid pairs become bi-oriented and separate towards opposite spindle poles through similar mechanisms as those observed in mitosis (*Schvarzstein et al., 2010*). In this process, centromeric cohesin plays a crucial role in establishing kinetochore geometry and thus determining sister kinetochore orientation (*Gryaznova et al., 2021*). In addition, the cohesin complex affects the formation of chromosome axial elements (AE) and the assembly of SC (*Agostinho et al., 2016*; *Fujiwara et al., 2020*; *Llano et al., 2012*). The cohesin complex is now emerging as a key player in mediating homolog bias of meiotic recombination (*Hong et al., 2013*; *Phipps and Dubrana, 2022*; *Sanchez-Moran et al., 2007*). Besides, the cohesin complex regulates gene expression in eukaryotes, acting as a transcriptional co-activator in humans that is recruited by sequence-specific transcription factors (*Casa et al., 2020*; *Kagey et al., 2010*; *Lara-Pezzi et al., 2004*; *Remeseiro et al., 2012*).

To date, the relationship between discrete cohesin subunits and their role in promoting sister chromatid cohesion remains elusive. In yeast, SCC1 ensures sister chromatid cohesin in mitosis and is associated with chromosomes during the S phase, dissociating during the metaphase-to-anaphase transition. The expression of SCC1 gradually decreases at the beginning of meiosis, suggesting a reduced role for SCC1 during meiosis (*Michaelis et al., 1997*). Instead, REC8, a specialized meiotic-specific cohesin component, replaces SCC1 in yeast (*Klein et al., 1999*; *Watanabe and Nurse, 1999*). REC8 also plays a crucial role in establishing sister kinetochore orientation in yeast, mice, and plants (*Chelysheva et al., 2005*; *Ogushi et al., 2021*; *Tóth et al., 2000*). Additionally, *rec8* mutants in flowering plants exhibit sticky chromosomes, which severely affects the localization of other key meiotic proteins (*Cai et al., 2003*; *Chelysheva et al., 2005*; *Shao et al., 2011*). SCC3 acts as an important component of the cohesin complex, which has been rarely studied in plants. The only research in *Arabidopsis* shows that SCC3 is responsible for maintaining centromeric orientation during meiosis (*Chelysheva et al., 2005*). However, the mechanisms through which SCC3 regulates the assembly of sister chromatid cohesion and its function in meiosis remain poorly understood. Whether SCC3 is similar to REC8 in regulating meiotic chromosome behavior in flowering plants remains unclear.

Here, we investigated the functional roles of SCC3 in rice, elucidating its participation in both mitosis and meiosis. Through our examination of chromosome 12 replication during interphase, we confirmed SCC3's role in stabilizing the chromosome structures, which is pivotal for sister chromatid cohesion in both mitosis and meiosis. Additionally, we explored the dynamics of cohesin loading and degradation on chromosomes using the antibody of SCC3 during mitosis and meiosis. SCC3 emerges as an axial element intricately involved in homologous pairing, synapsis, recombination progress, and crossover (CO) formation. Our findings also shed light on the regulation of SCC3 localization by REC8. Meanwhile, our investigation revealed the efficient repair of meiotic DSBs in *scc3*, utilizing sister chromatids as repair templates. We propose that SCC3 acts as a constituent of the cohesin complex, which plays an indispensable role in meiotic homologous pairing and synapsis.

## Results

### SCC3 causes both vegetative and reproductive growth defects

Considering SCC3 is a highly conserved protein in different species, we performed a BLAST search in the Rice Genome Annotation Project database (http://rice.plantbiology.msu.edu/cgibin/gbrowse/rice) and found a single hypothetical homolog of SCC3 proteins (*A. thaliana* SCC3, *M. musculus* STAG3 and *H. sapiens* SA1). This candidate protein is encoded by the *Os05g0188500* gene and shares the highest similarity with *AtSCC3*. The full-length cDNA of *SCC3* is 3755 nucleotides long and comprises 22 exons and 21 introns. SCC3 contains a 3345-nucleotide-long ORF that encodes an 1116 amino-acid protein. In higher eukaryotes, SCC3 is an evolutionary conserved protein that contains a highly conserved STAG domain (140-239aa), as predicted by SMART (*Figure 1—figure supplement 1A*). Multiple alignments of full-length SCC3 protein sequences and its homologs in other plants revealed the STAG domain is highly conserved across both monocotyledons and dicotyledons (*Figure 1—figure supplement 1B*). Homology modeling indicated that the structure of the SCC3 protein is strongly conserved among different species, which also possess similar STAG domains (*Figure 1—figure supplement 2A*). Additionally, a phylogenetic tree was constructed with full-length SCC3 amino-acid confirming that this protein is conserved in eukaryotes (*Figure 1—figure supplement 2B*). We also performed multiple sequence alignments of these proteins used in the evolutionary tree analysis (*Figure 1—figure supplement 3*).

A previous study in *Arabidopsis* (*Chelysheva et al., 2005*) identified a T-DNA insertion at the boundary between intron 5 and exon 6 of *SCC3* that caused lethality, whilst a weak allele was able to survive albeit with some developmental defects. To clarify the function of SCC3 in rice, we used the CRISPR-Cas9 system to generate three transgenic lines (*Figure 1—figure supplement 4*). The first allele (*scc3-1*) contains a frameshift 'GA' deletion in exon 2 and the second allele (*scc3-2*) contains a frameshift 'ACCGA' deletion in exon 11. We tested for allelism between these two *scc3* mutants by crossing *scc3-1$^{+/-}$* (male) and *scc3-2$^{+/-}$* (female). Of the 78 F$_1$ plants, 18 were *scc3-1$^{+/-}$*, 26 were *scc3-2$^{+/-}$*, 34 were wild-type for both loci. Thus, these two mutants are allelic and the heterozygous *scc3-1/scc3-2* is lethal. We were unable to isolate homozygous mutants in these two allelic progenies (n=128, 87 plants were *scc3-1$^{+/-}$* and 41 were wild-type; n=116, 80 plants were *scc3-2$^{+/-}$* and 36 were wild-type). These results suggest that *scc3-1* and *scc3-2* mutants are both embryo-lethal.

In addition, we obtained a weak transgenic line (*scc3*) containing a frameshift 'T' insertion in exon 19 leading to premature termination of translation. We found the *scc3* weak mutant exhibits abnormal vegetative growth (*Figure 1A and B*) and proceeded with further investigation of its other phenotypic effects. This revealed that *scc3* decreases plant height, tiller number, and panicle length (*Figure 1—figure supplement 5B–E*). To investigate the patterns of *SCC3* expression, we employed RT-PCR in various tissues and found the gene is ubiquitously expressed, particularly in roots, leaves, and panicles (*Figure 1—figure supplement 5A*). Accordingly, the weak mutant of SCC3 causes severe vegetative growth defects in rice plants.

To elucidate whether gametogenesis is normal in *scc3*, we performed cytological observation of anthers stained with 1% I$_2$-KI, and found almost all pollen grains were shrunken and inviable (*Figure 1D*). Our results also showed *scc3* flowers did not set seeds when pollinated with wild-type pollen, suggesting the mutant was both male and female sterile. We further observed heterozygous *scc3* plants produced progenies with a segregation ratio of 3:1 (normal: dwarf and sterile), indicating

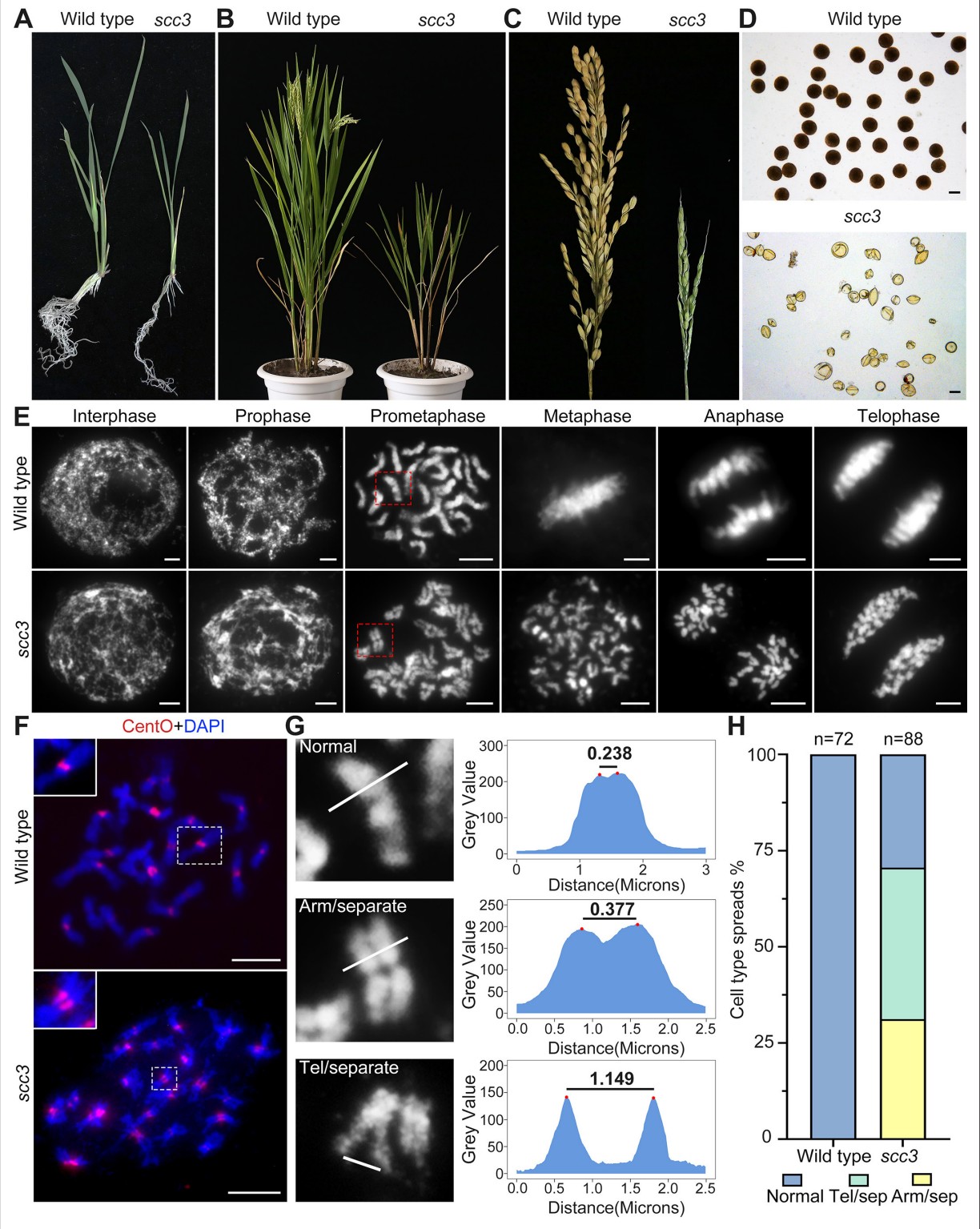

**Figure 1.** Sister chromatid cohesion 3 (SCC3) is required for sister chromatid cohesion during mitosis. (**A**) Morphology of seedlings and root tips in wild-type and *scc3*. (**B**) Morphology of plants in wild-type and *scc3*. (**C**) Morphology of panicles in wild-type and *scc3*. (**D**) Pollen grains stained in 1% I$_2$-KI solution in wild-type and *scc3*. Bars, 50 μm. (**E**) The chromosome behaviors of root tip cells in wild-type and *scc3*, stained with 4',6-diamidino-2-phenylindole (DAPI). Individual chromosomes are marked by red boxes. The distance between the sister chromatids increased significantly at prometaphase, and the sister chromatids separated in advance at metaphase in *scc3*. Bars, 5 μm. (**F**) Fluorescence in situ hybridization (FISH) analysis of mitotic cells with centromere-specific probes in the wild-type and *scc3*. The distance between centromeres increased significantly in *scc3*. CentO

*Figure 1 continued on next page*

*Figure 1 continued*

(red) signals indicate centromeres. Chromosomes were stained with DAPI (blue). Bars, 5 µm. (**G**) The distance between sister chromatids in different mitotic chromosomes. In normal condition, two sister chromatids form tight stick. In *scc3*, the distance between chromosome arms and telomeres were increased significantly. Curve diagrams show the distance between the arms and the telomeres, respectively, as measured by I$_{MAGE}$J. Bars, 1 µm. (**H**) Graphical representation of the frequency of each type of chromosome morphology. The classification was assigned when >50% chromosomes in a spread showed the indicated morphology.

The online version of this article includes the following figure supplement(s) for figure 1:

**Figure supplement 1.** Multiple sequence alignment of sister chromatid cohesion 3 (SCC3) with its homologs from three different species.

**Figure supplement 2.** Protein structure and phylogenetic tree analysis of sister chromatid cohesion 3 (SCC3).

**Figure supplement 3.** Multiple sequence alignment of sister chromatid cohesion 3 (SCC3) with its homologs from different species analyzed in an evolutionary tree.

**Figure supplement 4.** Schematic representation of sister chromatid cohesion 3s (SCC3's) mutation site.

**Figure supplement 5.** The expression pattern of sister chromatid cohesion 3 (SCC3) and plant phenotypic statistics of wild-type and *scc3* mutants.

the *scc3* mutation is recessive and monogenic. These data suggest that the *scc3* weak mutation dramatically interferes with both plant vegetative and reproductive growth.

## SCC3 is required for sister chromatid cohesion during mitosis

To determine whether SCC3 is involved in sister chromatid cohesion during mitosis, we observed chromosome behavior in root tip cells of *scc3* and wild-type plants. In wild-type plants, DNA replicates and folds into an ordered structure during interphase (*Figure 1E*). Subsequently, chromatin condenses to form rod-like chromosomes during prophase, further aligning at the equatorial plate during metaphase before segregating to distinct poles of the cell.

In contrast, *scc3* exhibited 'X'-shaped chromosomes with distant arms and telomeres at prometaphase (*Figure 1E and G*), indicating compromised cohesion between sister chromatids. Notably, approximately 70.6% of *scc3* root tip cells (n=88) exhibited partially or completely separated sister chromatids (*Figure 1H*). During the transition from prometaphase to metaphase, all sister chromatids prematurely separated and failed to align properly at the equatorial plate. From anaphase to telophase, sister chromatids were loosely pulled towards the spindle poles (*Figure 1E*). However, aneuploidy was not observed after anaphase, as mitotic tubulin remained attached to the centromeres of sister chromatids at metaphase, ensuring their equal segregation.

Additionally, we monitored the distance between centromeres using FISH probes with the centromere-specific tandem repetitive sequence CentO in both wild-type and *scc3* somatic cells during prometaphase. We observed a substantial increase in the distance between sister centromeres in *scc3* (*Figure 1F*, n=25). However, despite the distance between sister chromatids increased in *scc3*, sisters were still observed side-by-side during prometaphase, suggesting the presence of other cohesion proteins maintaining sister chromatid association. Our results demonstrated that SCC3 is required for cohesion between sister arms and centromeres during mitosis.

To elucidate the underlying cause of the increased distance between sister chromatids, we performed full-length FISH assays to monitor the dynamic phenotype of chromosome 12 during interphase. In the wild-type, replicating sister chromatids were observable during interphase (*Figure 2A*). Subsequently, two chromosome clumps formed during prophase and developed into high-order linear chromosomes during prometaphase. During this period, the two sister chromatids overlapped and condensed into a short stick. However, in *scc3*, no chromosome clumps were detected during interphase, and an evident uncondensed chromosome structure was observed, suggesting a role for SCC3 in organizing genome topology and chromatin association dynamics. During prophase, chromosome 12 remained in a loose lump and was unable to condense into a stick-like structure. At prometaphase, the abnormal chromatin finally formed loosely connected sister chromatids (*Figure 2A*). These observations implicate SCC3 involvement in the dynamic structural change of chromatin during interphase, influencing the ability of cohesin to bind sister chromatids, potentially regulating chromatin composition, and genome dynamics.

To investigate the localization of SCC3 during mitosis, we generated a mouse polyclonal antibody against the C-terminal of SCC3 and performed immunostaining in wild-type root tip cells (*Figure 2B*). No SCC3 signal was detected in *scc3* root tip cells, confirming the specificity of the SCC3 antibody

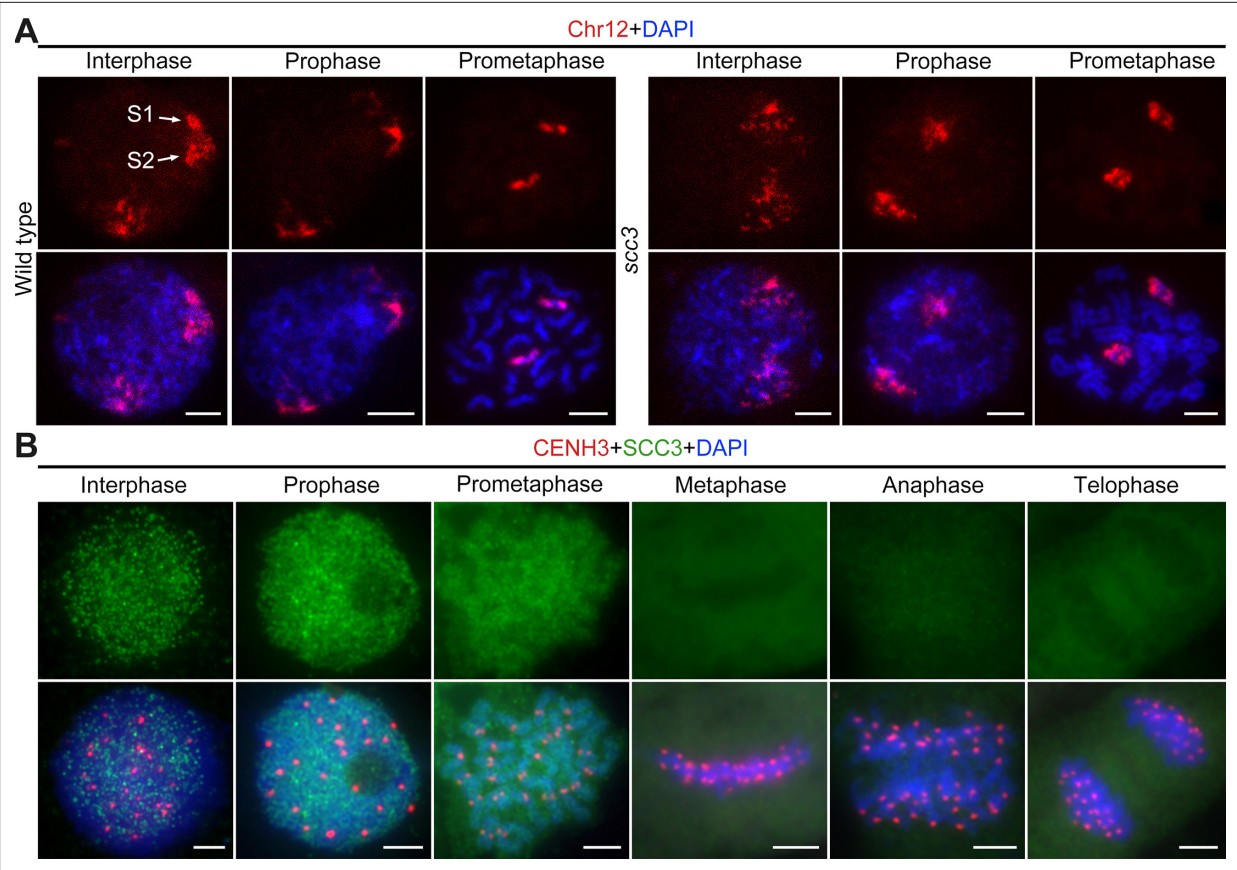

**Figure 2.** Sister chromatid cohesion 3 (SCC3) alters the structure of sister chromatids during mitosis. (**A**) The dynamic process of chromosome structure in early mitosis as revealed by pooled oligos specific to chromosome 12 (red). In the wild-type, S1 and S2 indicate the replicated sister chromatids. However, in *scc3* sister chromatids exhibited the variation of structure from interphase to prophase. Mitotic chromosomes in wild-type and *scc3* were stained with 4',6-diamidino-2-phenylindole (DAPI) (blue). Bars, 5 μm. (**B**) The loading pattern of SCC3 (green, from mouse) and CENH3 (red, from rabbit) in wild-type root tip cells. Chromosomes were stained with DAPI (blue). Bars, 5 μm.

The online version of this article includes the following figure supplement(s) for figure 2:

**Figure supplement 1.** Immunolocalization of sister chromatid cohesion 3 (SCC3) in *scc3* mitosis and mitosis.

utilized in mitosis (*Figure 2—figure supplement 1A*). In most instances, SCC3 was uniformly distributed within the nuclear during interphase. As cell division started, SCC3 immunosignals around centromeric regions intensified at prophase. SCC3 eventually coalesced into intact chromosomes at prometaphase. From metaphase to telophase, SCC3 signals completely disappeared (*Figure 2B*). These observations suggest that SCC3 may enhance the cohesin-DNA interaction during the transition from interphase to prophase and gradually dissociates from chromosomes after prometaphase.

## SCC3 acts as an axial element during meiosis

To evaluate SCC3 localization during meiosis, we performed dual-immunolocalization between SCC3 and REC8 in wild-type meiocytes. REC8 is a specific axial protein specific to meiosis (*Shao et al., 2011*). REC8 signals were initially detected as foci in the nucleus at leptotene, during which SCC3 was simultaneously detectable at the same time (*Figure 3A*). From leptotene to zygotene, SCC3 signals were detected as similar elongated foci that formed filamentous structures on the chromosomes. During pachytene and diplotene, constant SCC3 signals were distributed along the chromosome axes (*Figure 3A*). At diakinesis, axial SCC3 signals became diffuse and centromeric signals gradually became clear. At metaphase I, only centromeric SCC3 signals remained (*Figure 3B*). However, REC8 signals were difficult to be detected after diakinesis (*Figure 3—figure supplement 1*). In order to further clarify the localization pattern between SCC3 and REC8, we performed super-resolution imaging on wild-type meiocytes. We found REC8 and SCC3 signals were parallel and colocalized

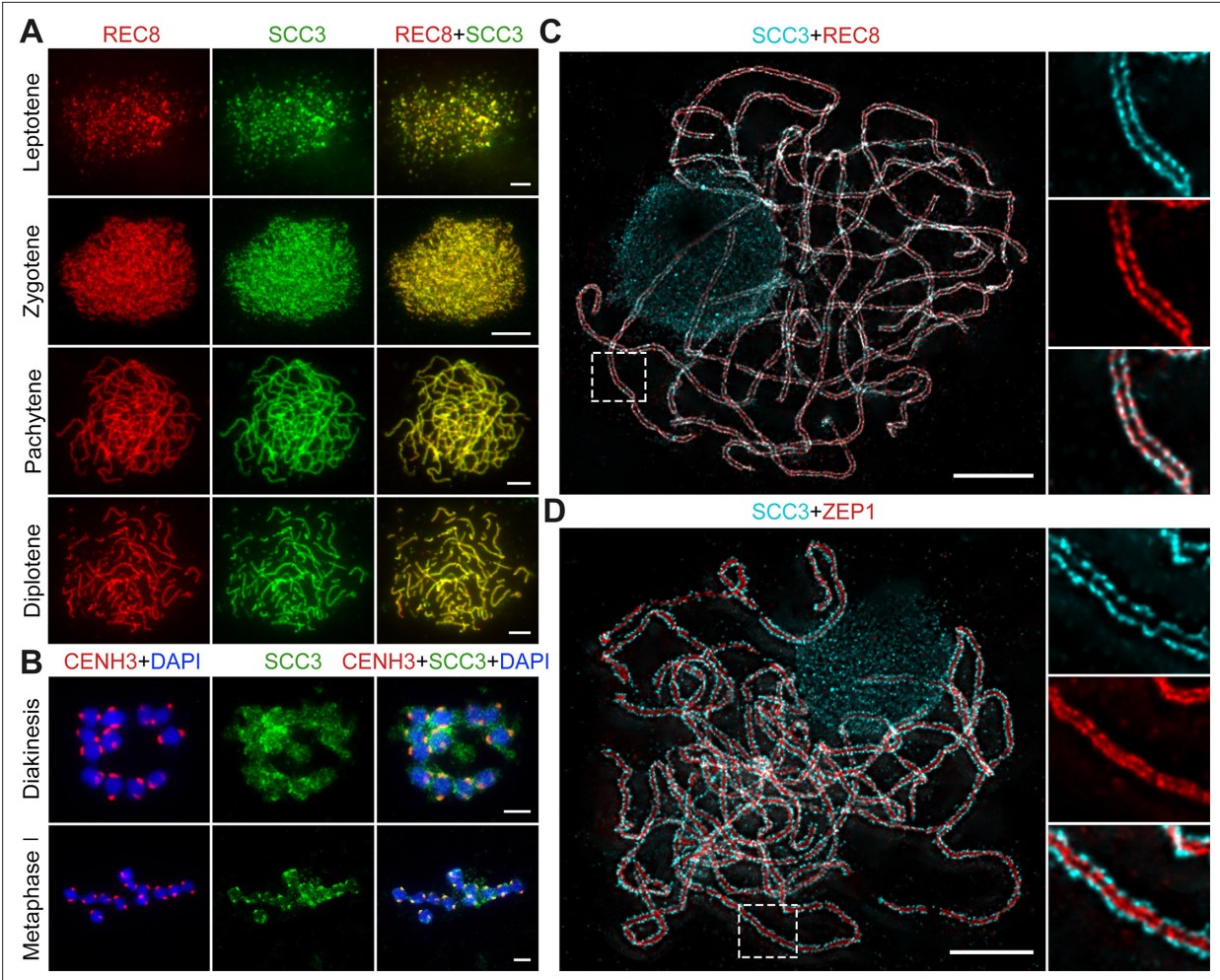

**Figure 3.** Sister chromatid cohesion 3 (SCC3) is an axial element during meiosis. (**A**) In meiosis, SCC3 (green, from mouse) colocalizes with REC8 (red, from rabbit) from the leptotene to diplotene. Bars, 5 μm. (**B**) During diakinesis and metaphase I, SCC3 (green, from mouse) gradually dispersed and finally retained in the centromeres indicated by CENH3 (red, from rabbit). Chromosomes were stained with 4',6-diamidino-2-phenylindole (DAPI) (blue). Bars, 5 μm. (**C**) Immunostaining of SCC3 (cyan, from mouse) and REC8 (red, from rabbit) in wild-type meiocytes at pachytene. Two parallel linear SCC3 signals colocalize with the REC8 linear signals, indicating chromosomal axial elements. Magnified images of the blocked regions are shown on the right. Bars, 5 μm. (**D**) Immunostaining of SCC3 (cyan, from mouse) and ZEP1 (red, from rabbit) in wild-type meiocytes at pachytene. Two linear SCC3 signals wrap the ZEP1 signals, indicating central elements of SC are wrapped by axial elements. Magnified images of the blocked regions are shown on the right. Bars, 5 μm.

The online version of this article includes the following figure supplement(s) for figure 3:

**Figure supplement 1.** Immunolocalization of CENH3 and REC8 in wild-type from diakinesis to metaphase I.

forming the parallel axis along chromosome length (*Figure 3C*). In addition, the central element ZEP1 formed two parallel lines surrounded by SCC3 signals (*Figure 3D*). No SCC3 signal was present in *scc3* meiocytes, confirming the specificity of the SCC3 antibody utilized in immunodetection (*Figure 2— figure supplement 1B*).

## SCC3 is required for sister chromatids cohesion in early meiosis

The premeiotic interphase bears the resemblance to mitosis, wherein sister chromatids undergo replication and are tethered together by cohesins. To determine whether SCC3 is involved in sister chromatid cohesion during meiosis, we scrutinized the chromosome dynamics of meiocytes in both *scc3* and wild-type before leptotene. Detailed cytological phenotypes of meiotic initiation in rice have been previously elucidated (*Zhao et al., 2018*). In the wild-type, the initial discernible stage of meiosis is preleptotene, during which sister chromatids have completed their replication. At this stage, each

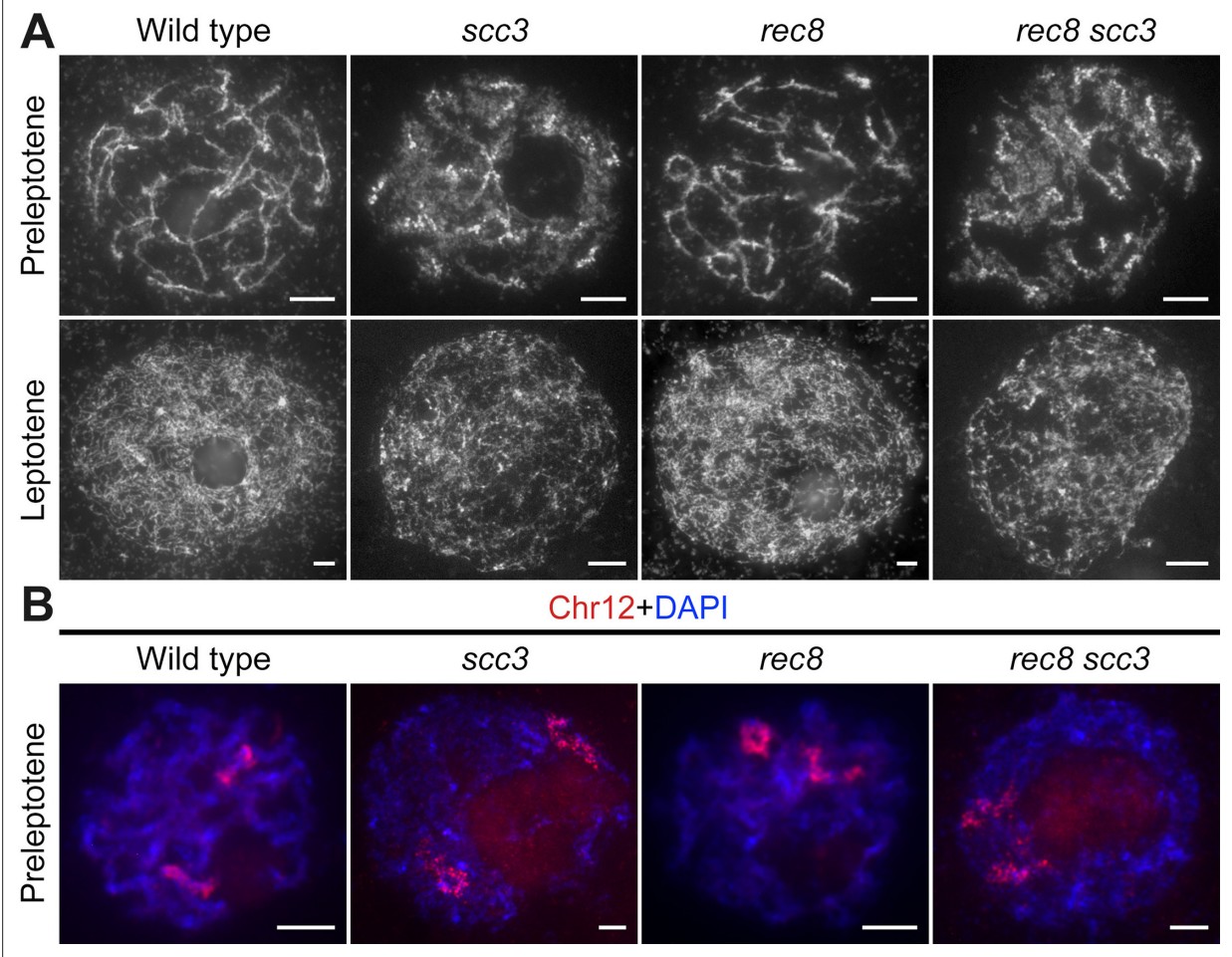

**Figure 4.** Sister chromatid cohesion 3 (SCC3) is required for sister chromatid cohesion in early meiosis. (**A**) Meiocyte chromosome morphology at the preleptotene and leptotene in wild-type, *scc3*, *rec8*, and *rec8 scc3*. Chromosomes at preleptotene exhibited a hairy rodlike appearance and formed thin threads at leptotene in wild-type. However, chromosomes in *scc3* were dispersed with blurred outlines and failed to form thin threads at leptotene. Bars, 5 μm. (**B**) The dynamic process of chromosome structure in preleptotene was revealed by pooled oligos specific to chromosome 12 (red). Preleptotene chromosomes in wild-type, *scc3*, *rec8*, and *rec8 scc3*. Chromosomes were stained with 4',6-diamidino-2-phenylindole (DAPI) (blue). Bars, 5 μm.

chromosome exhibited a condensed configuration, similar to the mitotic prophase chromosome (*Figure 4A*). Subsequently, they turned into fine threads at leptotene. However, in *scc3*, chromosomes showed defused morphology without clear outlines at preleptotene. These observations suggest that SCC3 affects the morphological construction of early meiotic chromosomes.

We also scrutinized the chromosome behavior in both the *rec8* single mutant and the *scc3 rec8* double mutant at preleptotene (*Figure 4A*). Surprisingly, *rec8* exhibited elongated chromosomes similar to the wild-type, whereas *scc3 rec8* displayed diffused chromosomes same as *scc3*. These results suggest that REC8 does not impede the early replication process of sister chromatids, and the chromosomal abnormalities observed in the *scc3 rec8* double mutant would be attributed to the SCC3 mutation. We further performed full-length FISH assays to monitor chromosome 12 at preleptotene (*Figure 4B*). The labeled chromosome 12 formed short rod-shaped threads in both wild-type and *rec8*, whereas it appeared diffuse structure in both *scc3* and *scc3 rec8*. These data together with those from DAPI staining, further prove that SCC3 has an important function on chromosome structure during early meiosis, which is independent of REC8.

### Homologous chromosome pairing and synapsis are disturbed in *scc3*

To elucidate the causes of sterility in *scc3*, we investigated the male meiotic chromosome behavior of wild-type and *scc3* meiocytes. In wild-type meiocytes, homologous chromosomes initiated pairing

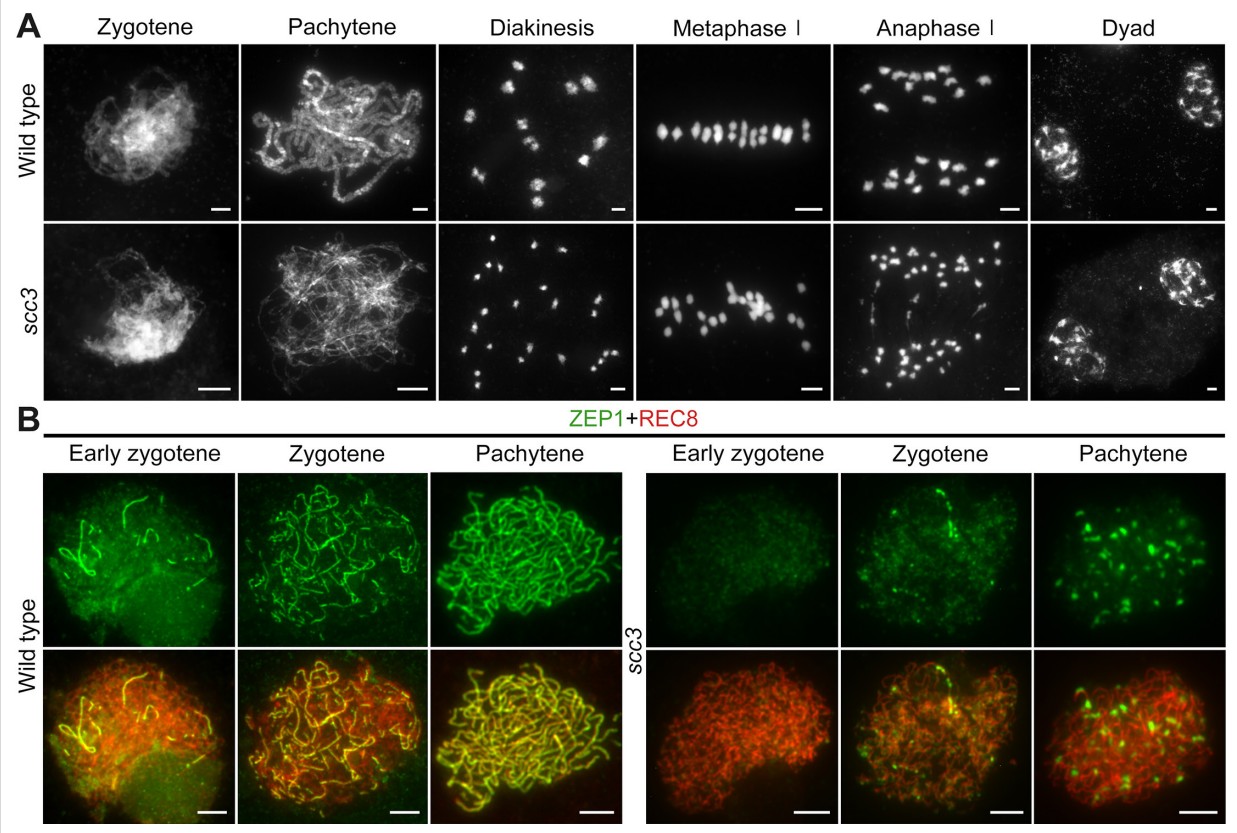

**Figure 5.** Homologous pairing and synapsis are disturbed in sister chromatid cohesion 3 (*scc3*). (**A**) Meiotic chromosome behavior in wild-type and *scc3*. Bars, 5 μm. (**B**) Immunolocalization of ZEP1 (green, from mouse) and REC8 (red, from rabbit) in wild-type and *scc3* meiocytes. ZEP1 was severely suppressed from early zygotene to pachytene in *scc3*. Bars, 5 μm.

The online version of this article includes the following figure supplement(s) for figure 5:

**Figure supplement 1.** Chromosome behaviors of *spo11* and cytogenetic analysis of homologous pairing in wild-type and sister chromatid cohesion 3 (*scc3*).

**Figure supplement 2.** Immunolocalization of ZEP1, PAIR2, and PAIR3 in sister chromatid cohesion 3 (*scc3*) at zygotene.

during the early zygotene stage and completed synapsis at pachytene. Additionally, at metaphase I, 12 bivalents were methodically arranged on the equatorial plate, facilitating the segregation of homologous chromosomes during anaphase I (*Figure 5A*). The chromosomal dynamics in *scc3* did not deviate from the wild-type during zygotene. However, the onset of meiotic anomalies became evident at pachytene, characterized by the dispersion of most chromosomes as solitary filaments due to inadequate homologous pairing (*Figure 5A*). Twenty-four univalents were observed from diakinesis to metaphase I (n=161, 150 meiocytes had 24 univalents, while 11 meiocytes had 1–2 bivalents). Two sister chromatids of each univalent were always pulled to opposite poles leading to 48 sister chromatids being observed at anaphase I. Notably, in *spo11-1* mutants implicated in the disruption of meiotic DSB formation and similarly exhibiting 24 univalents as observed in *scc3*, the separation of sister chromatids occurred either haphazardly or not at all (*Figure 5—figure supplement 1A*). These findings indicate that the mutation of SCC3 will lead to premature separation of sister chromatids at anaphase I.

The formation of the telomere bouquet constitutes an essential precondition for homologous chromosome pairing during early zygotene (*Zhang et al., 2020*). To ascertain the occurrence of bouquet formation, we employed fluorescence in situ hybridization (FISH) assays targeting telomeres with a specific probe (pAtT4) in both wild-type and *scc3*. In the wild-type, telomeres congregated to a specified region, culminating in the formation of a quintessential bouquet at the nuclear envelope during zygotene. Notably, *scc3* mutants also demonstrated typical telomere bouquet formation (*Figure 5—figure supplement 1B*). Subsequently, we utilized 5 S ribosomal DNA (5 S rDNA) as a marker to

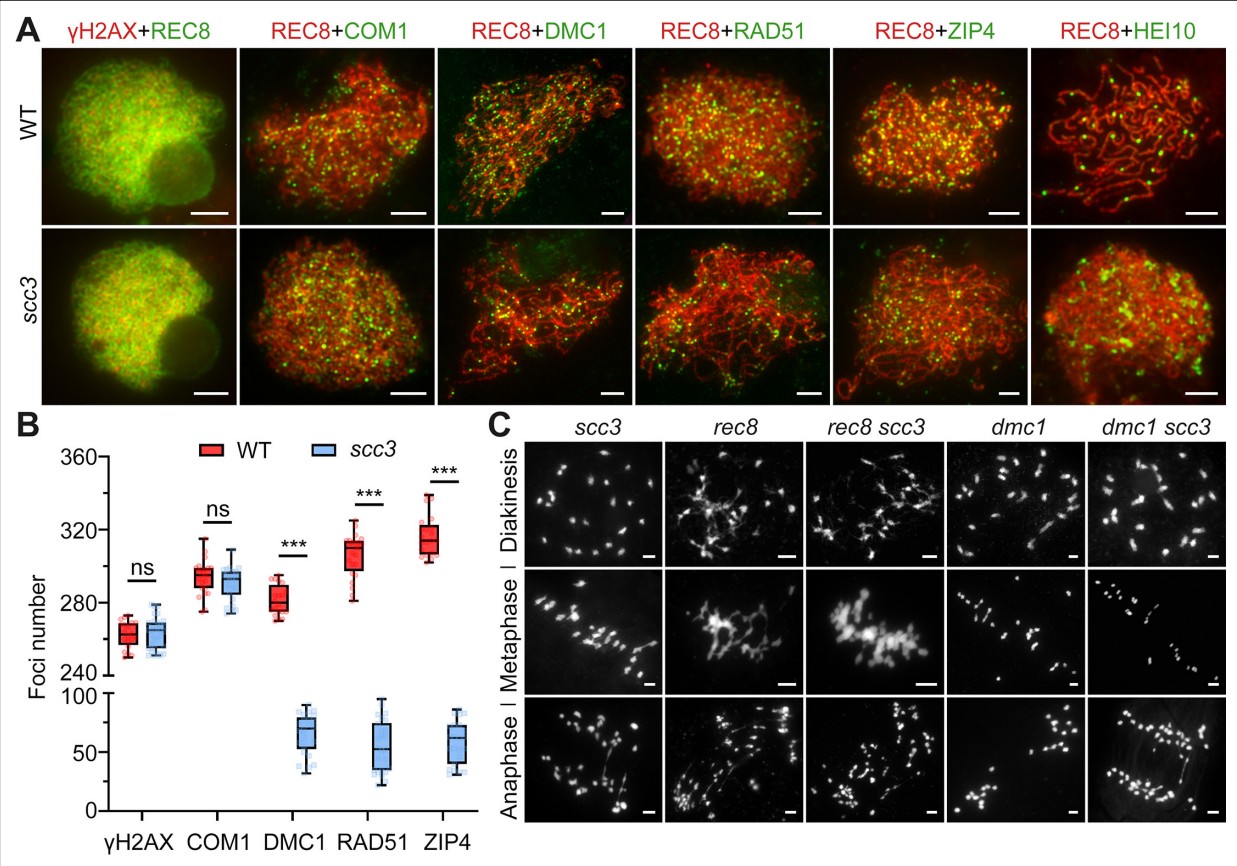

**Figure 6.** Sister chromatid cohesion 3 (SCC3) affects recombination progress and crossover (CO) formation. (**A**) Immunolocalization of histone γH2AX (red), COM1 (green, from mouse), DMC1 (green, from mouse), RAD51 (green, from rabbit), ZIP4 (green, from mouse), and HEI10 (green, from mouse) in wild-type and *scc3*. REC8 (red, from mouse and rabbit) signals were used to visualize chromosome axes. Bars, 5 µm. (**B**) Box scatter plot of histone γH2AX phosphorylation, COM1, DMC1, RAD51, and ZIP4 in wild-type and *scc3*. No difference of histone γH2AX and COM1 were shown between the wild-type and *scc3*. DMC1, RAD51, and ZIP4 foci were significantly decreased in *scc3* compared with wild-type. Values are means ± SD. \*\*\* represents p<0.001, two-tailed Student's *t*-tests was performed. (**C**) Chromosome behaviors in *scc3*, *rec8*, *rec8 scc3*, *dmc1*, and *dmc1 scc3* from diakinesis to anaphase I. Bars, 5 µm.

evaluate homologous pairing. One overlapping 5 S rDNA focus was observed in wild-type pachytene, but two separate foci were observed in *scc3*, indicative of unpaired homologous chromosomes (n=52; *Figure 2—figure supplement 1C*). This observation corroborates the function of SCC3 in facilitating homologous chromosome pairing.

Homologous chromosomes align and are subsequently juxtaposed via a proteinaceous structure known as the synaptonemal complex (SC), which comprises two parallel arrays of axial elements (AE), termed lateral elements (LE) prior to synapsis. These elements lay the groundwork for SC assembly (*Ren et al., 2021*). Given SCC3's identification as a component of the AE, we examined the SC assembly process by immunostaining with an antibody against ZEP1 in both *scc3* and wild-type meiocytes. In the wild-type, ZEP1 progressively elongated, forming brief linear signals during zygotene that fully extended along the paired homologous chromosomes by pachytene (*Figure 5B*). Conversely, *scc3* mutants exhibited ZEP1 as isolated punctate foci at early zygotene, a pattern that persisted into pachytene (*Figure 5B*), underscoring the critical role of SCC3 in SC assembly. Furthermore, the localization of other AE components (PAIR2 and PAIR3) was also investigated, and these proteins displayed normal behavior in *scc3* (*Figure 5—figure supplement 2*).

### SCC3 is essential for recombination progress and CO formation

Since homologous pairing was impaired in *scc3* meiocytes, we wondered whether recombination was normal in *scc3*. γH2AX is a marker that is used as a proxy to assess DSB formation. Immunostaining

results revealed no significant difference in the quantity of γH2AX foci between the wild-type and *scc3* (265.4±8.6, n=31 for *scc3* meiocytes; 262.2±7.2, n=20 for wild-type meiocytes) (*Figure 6A*). Furthermore, we examined the DSB resection factor (COM1), meiotic strand-invasion and exchange factors (RAD51, DMC1) and the interference-sensitive crossover (CO) marker (ZIP4) in both wild-type and *scc3*. Immunostaining results revealed that the number of COM1 foci (290.3±8.8, n=26 for *scc3* meiocytes; 293.8±9.4, n=23 for wild-type meiocytes) showed no significant difference between the wild-type and *scc3* (*Figure 6A and B*). However, a significant reduction in the foci of DMC1 (65.4±17.1, n=25 for *scc3* meiocytes; 281.2±8.3, n=20 for wild-type meiocytes), RAD51 (55±21.7, n=26 for *scc3* meiocytes; 306±11.9; n=24 for wild-type meiocytes), and ZIP4 (58±17.9, n=23 for *scc3* meiocytes; 316.6±11.1, n=21 for wild-type meiocytes) was observed in *scc3*, suggesting a disruption in recombination progression and CO formation. Previous studies have identified HEI10 as a marker of Class I COs. In the wild-type, prominent HEI10 foci were observed (24.2±3.8, n=25) at late pachytene (*Figure 6A and B*). Conversely, *scc3* predominantly exhibited sparse HEI10 foci, lacking in prominent, bright signals. Collectively, these results suggest that CO maturation was disturbed in *scc3*. Although a minor portion of recombinant proteins could be localized, the homologous recombination process was still greatly restricted.

## Meiotic DSBs are repaired using sister chromatids in *scc3*

It is noteworthy that the formation of DSBs appears normal in *scc3* (*Figure 6A*). However, the destabilization of homologous pairing and synapsis in *scc3* leads to aberrations in homologous recombination. Despite the absence of the homologous recombination process, 24 univalents were observed from diakinesis to metaphase I. This was coupled with the complete separation of sister chromatids during anaphase I without chromosome fragmentation, suggesting proficient repair of these meiotic DSBs in *scc3*. These observations indicate that the meiotic DSBs in *scc3* most likely utilize sister chromatids as repair templates. Additionally, we examined the chromosome behavior of both *rec8* and *rec8 scc3* from diakinesis to anaphase I (*Figure 6C*). They all exhibited abnormal chromosome morphology, manifesting a highly adhesive phenotype from diakinesis to metaphase I. Furthermore, chromosome fragmentation was observed in anaphase I, indicating inadequate repair of meiotic DSBs in both *rec8* and *rec8 scc3*. Moreover, DMC1 has been implicated in inter-sister exchange during meiosis in both *Arabidopsis* and rice (*Kurzbauer et al., 2012*; *Wang et al., 2016*). To gain a deeper understanding of the relationship between DMC1 and SCC3 in rice, we generated the *dmc1 scc3* double mutant and monitored the meiosis process from diakinesis to anaphase I. The meiotic defects of *dmc1* and *dmc1 scc3* were similar from diakinesis to metaphase I, resulting in 24 univalents (*Figure 6C*). However, the sister chromatids of *dmc1 scc3* were completely separated in anaphase I, which was consistent with the phenotype of *scc3*. This suggests that these two genes might function independently in sister-chromatids repair during meiosis.

## The loading of SCC3 onto meiotic chromosomes depends on REC8

After establishing SCC3's role as an axial element in meiosis, an assessment was conducted to ascertain whether its localization was influenced by other meiotic AEs. REC8, PAIR2, and PAIR3 are known to localize on chromosome axes in rice, playing pivotal roles in facilitating SC assembly. The distribution of SCC3 signals remained unaffected in *pair2* and *pair3* mutants during meiosis (*Figure 7B*). Moreover, the distribution of SCC3 was normal in other mutants that affect meiotic DSB formation (*spo11-1*), strand invasion and homology searching (*com1* and *dmc1*), and synaptonemal complex formation (*zep1*) (*Figure 7B*). This indicates that SCC3 localization is independent of the recombination process. However, a notable absence of SCC3 signal distribution was observed in *rec8*, indicating that REC8 plays a crucial role in the normal loading progression of SCC3 (n=36). Immunostaining of γH2AX confirmed normal DSB formation in *rec8* (*Figure 7A*). In addition, the localization of the DSB end resection factor COM1 (n=25) and the inter-homologous strand-invasion factor RAD51 (n=26) was impaired in *rec8* (*Figure 7A*). Furthermore, PAIR2 (n=28) only form short, interrupted stretches, and few detectable PAIR3 signals were observed (n=32). These findings indicate that meiotic DSB formation remains unaffected in *rec8*, while the processes of DSB end resection and homologous recombination are significantly compromised.

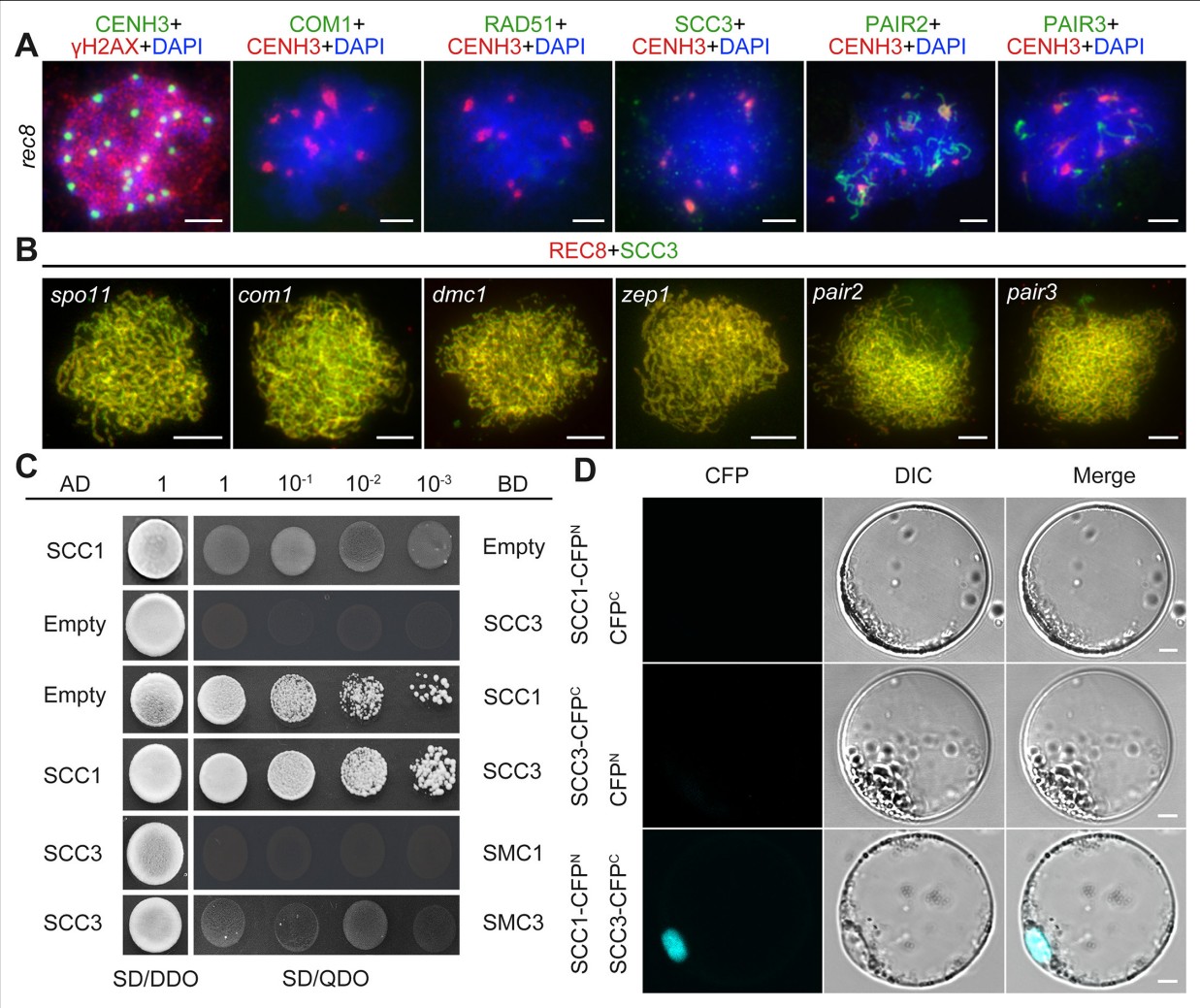

**Figure 7.** Sister chromatid cohesion 3 (SCC3) loading onto meiotic chromosomes depends on REC8. (**A**) Immunolocalization of γH2AX (red, from rabbit), COM1 (green, from mouse), RAD51 (green, from rabbit), SCC3 (green, from mouse), PAIR2 (green, from rabbit), and PAIR3 (green, from rabbit) in *rec8* meiocytes at zygotene. CENH3 (red and green, from rabbit and mouse) was used to indicate the centromeres. Bars, 5 µm. (**B**) Immunolocalization of REC8 (red, from rabbit) and SCC3 (green, from mouse) in *spo11*, *com1*, *dmc1*, *zep1*, *pair2*, and *pair3* meiocytes at zygotene. Bars, 5 µm. (**C**) SCC3 interacts with RAD21.1 in yeast-two-hybrid assays. Interactions between bait and prey were examined on SD/DDO (SD-Leu-Trp) control media and SD/QDO (SD-Ade-His-Leu-Trp) selective media. AD, prey vector; BD, bait vector. (**D**) Bimolecular fluorescence complementation assays show the interactions between SCC3 and RAD21.1 in rice protoplasts. Bars, 5 µm.

The online version of this article includes the following figure supplement(s) for figure 7:

**Figure supplement 1.** Sister chromatid cohesion 3 (SCC3) does not interact with REC8, SMC1, or SMC3.

**Figure supplement 2.** The expression levels of cohesin-related genes in wild-type and other mutants.

**Figure supplement 3.** Sister chromatid cohesion 3 (SCC3) acts as the cohesin and promotes homologous pairing and synapsis.

## SCC3 interacts with SCC1

The cohesin complex is a ring-shaped protein complex where subunits interact with each other (*Li et al., 2018a*; *Murayama and Uhlmann, 2015*; *Roig et al., 2014*). To clarify the relationship between different subunits in rice, we performed Y2H assays between core cohesin proteins. Surprisingly, we found that SCC3 only interacts with SCC1 (*Figure 7C*), which was further confirmed by BiFC assays (*Figure 7D*). However, it is worth noting that SCC3 could not interact with other cohesin proteins, including REC8, SMC1, and SMC3 (*Figure 7—figure supplement 1*), indicating that SCC3 could function directly in combination with SCC1 in the cohesin complex.

Previous studies have shown that cohesins regulate gene expression (*Cuadrado et al., 2015*; *Gligoris and Löwe, 2016*; *Remeseiro et al., 2012*). Therefore, we investigated the expression levels of cohesin-related genes in wild-type and *scc3* spikelets using RT-qPCR. Interestingly, we found that the cohesin subunits *SCC1* and *REC8*, as well as the centromeric cohesion protector *SGO1*, were significantly downregulated in *scc3* (*Figure 7—figure supplement 2A*). In contrast, the cohesin subunits *SMC1* and *SMC3*, as well as the lateral/central elements *PAIR3* and *ZEP1*, showed no differences in gene expression. The only upregulated axial element in *scc3* spikelets was *PAIR2*.

We then investigated the expression levels of cohesin genes in axial/lateral element mutants (*pair2* and *pair3*) (*Figure 7—figure supplement 2B*), and found that *SMC3*, *SCC3,* and *SCC1* were significantly downregulated compared to the wild-type. These findings suggest that SCC3 may thus be involved in the transcriptional regulation of meiosis genes, while axial elements affect cohesin proteins by regulating the expression levels of associated genes.

## Discussion

Chromosome segregation during mitosis and meiosis encompasses a series of meticulously orchestrated events, including the formation of the cohesin complex, which serves diverse functions throughout the cell cycle (*Moronta-Gines et al., 2019*; *Nasmyth and Haering, 2009*). Here, we show that SCC3 is a conserved cohesin subunit affecting sister chromatid assembly in both mitosis and meiosis. Crucially, our results also indicate that SCC3 is a component of LE/AE that is required for homologous pairing and synapsis during meiosis, which is essential for the recombination process and CO formation.

Most research on the cohesin complex predominantly emphasizes its role during mitosis, attributed to the challenges associated with observing the morphology of sister chromatids throughout meiosis. In human cells, the separation of sister chromatids during mitosis occurs in the absence of the *SCC3* homologs, *SA1* and *SA2*. SA1 is indispensable for the cohesion of telomeres and chromosomal arms, while SA2 is crucial for centromeric cohesion (*Canudas and Smith, 2009*). In rice, *SCC3* has only a single copy, therefore, we observed all kinds of multifaceted defects in *scc3*. SCC3 is paramount not only for the stability of sister chromatid's structure but also for facilitating the loading of cohesin. In yeast, the cohesin complex comprising SCC1, SMC1, and SMC3 remains intact in *scc3*. However, its chromosomal localization is impeded, most likely due to SCC3's role in maintaining the stability of cohesion (*Orgil et al., 2015*; *Pathania et al., 2021*). Furthermore, SCC3 harbors a conserved DNA binding domain in the SCC3-SCC1 subcomplex, essential for the chromatin association of cohesins (*Higashi et al., 2020*). In yeast, the DNA-binding domain of the SCC3-SCC1 facilitates the recruitment of cohesin complexes to chromosomes, thus enhancing cohesin's functionality during mitotic cell division (*Kulemzina et al., 2012*; *Li et al., 2018a*). These findings suggest that the interaction between SCC3 and SCC1 is vital for the stability of the cohesin ring and its continued association with DNA. In *Arabidopsis*, SCC3 is also involved in vegetative growth of plants, but its role in mitosis remains unclear (*Chelysheva et al., 2005*). Leveraging these antecedent insights, our investigation corroborates that SCC3 is integral to the initial formation of cohesion during mitosis. In the rice *scc3* mutant, vegetative growth was severely inhibited. Our analyses indicate an aberration in the chromosomal structure during interphase in *scc3* mutants, possibly due to SCC3's necessity for stable chromosomal cohesin binding. Although an augmentation in the distance between sister chromatids was observed during prometaphase, the sister chromatids remained partially dissociated, suggesting the residual functionality of other cohesin subunits. Collectively, our findings underscore SCC3 as a conserved and pivotal cohesin protein within the mitotic process.

SCC3 not only functions in mitosis but also affects meiotic chromosome structure and function. In vertebrates, *SCC3* is encoded by two paralogous genes, including the meiosis-specific subunit *STAG3*. Investigations utilizing SIM analysis on human *stag3* have elucidated that chromosomal axes are regionally dissociated, and there is a substantial reduction in the synaptonemal complex (*Biswas et al., 2016*; *Fukuda et al., 2014*; *Hopkins et al., 2014*; *Llano et al., 2012*). These observations suggest that STAG3 is required for synapsis between homologous chromosomes. In comparison, the morphological characteristics of meiocytes during meiosis are more discernible in plant cells. Specifically, in *Arabidopsis scc3*, a complete absence of fertility has been observed, underscoring the pivotal role of SCC3 in meiotic process, including unstable homologous pairing at the pachytene (*Chelysheva et al., 2005*). In analogous studies with other cohesin mutants, *Atscc2* has been identified

as essential for the recruitment of the meiosis-specific cohesin subunit REC8 (*Wang et al., 2020*). This suggests that mutations in cohesin-associated genes could potentially compromise the cohesin complex's chromosome-binding capacity. Accordingly, in our investigation on meiosis, it has been determined that rice SCC3 facilitates the maintenance of the structure of sister chromatids during the preleptotene. More importantly, SCC3 is localized between homologous chromosomes as a critical AE, ensuring accurate homologous pairing and synapsis. SCC3's presence is crucial for the normal localization of DMC1, RAD51, ZIP4, and HEI10, elements vital for progression of recombination and the formation of CO. Our proposed model suggests that SCC3 is a part of the cohesin ring, and functions in specific association with SCC1 (*Figure 7—figure supplement 3A*). Given SCC3's multi-faceted roles in cohesion dynamics and meiotic processes, we hypothesize that SCC3 may function as an axial element to stabilize loop boundaries in meiosis, thereby promoting the homologous pairing and synapsis (*Figure 7—figure supplement 3B*). Notably, it has been proposed that the facilitation of meiotic inter-homolog recombination is attributed to an inhibition of inter-sister-chromatid recombination, which is enforced by meiosis-specific constituents of the chromosome axis (*Goldfarb and Lichten, 2010*). In *scc3* mutants, meiotic DSBs seemed to utilize sister chromatids as repair templates exclusively. Although essential meiotic proteins such as COM1, DMC1, and RAD51 were decreased in *scc3*, these proteins may still function in facilitating the inter-sister chromatid repair pathway, as DMC1 and RAD51 are also involved in the repair process of inter-sister chromatids. We postulate that the presence of cohesin predisposes meiotic recombination towards an inter-homolog bias for repair. Conversely, in the absence of cohesin and when homologous recombination is compromised, recombination is biased towards inter-sister repair.

Cohesin is a multi-protein complex that plays an indispensable role in establishing sister chromatid cohesion and the high-order chromosome architecture in both somatic and germ cells. Distinct meiosis-specific subunits likely execute divergent functions (*Cuadrado et al., 2015*; *Ishiguro, 2019*; *Schvarzstein et al., 2010*). Our findings suggest that SCC3 and REC8 assume differential roles during mitosis and meiosis, respectively. REC8 was identified as a meiosis-specific protein, assuming the role of SCC1 in meiosis to constitute the cohesin complex. Our investigation reveals that although REC8 and SCC3 exhibit similar localization patterns, but they show many differences. Significantly, REC8 is completely absent in mitotic cells. Conversely, our data indicate that SCC3 is operative in both mitosis and meiosis, potentially playing a role in the loading of cohesins onto sister chromatids during interphase. It is noteworthy that the cohesin loading mechanism onto chromosomes during mitosis and meiosis remains consistent. Meiotic cohesin is loaded onto chromosomes during interphase, likely coinciding with the replication and meiotic fate acquisition of sporogenous cells. This indicates that REC8 does not participate in cohesion establishment at this juncture. Furthermore, we observed SCC3 signals at centromeres from diakinesis to metaphase I, whereas REC8 was undetectable at metaphase I. These distinct localization patterns further imply they may have divergent functional roles in meiosis. Previous research on *Arabidopsis rec8* mutants highlighted chromosomes exhibiting a sticky phenotype that remained challenging to elucidate (*Cai et al., 2003*; *Lambing et al., 2020*). Similarly, rice *rec8* meiocytes exhibited defective chromosomes with a pronounced sticky phenotype from diakinesis to metaphase I (*Shao et al., 2011*). In our study, chromosomes in *scc3* did not display the stickiness observed in *rec8* (*Figure 6C*). Moreover, the majority of meiotic proteins (including SCC3) failed to localize in *rec8*. This indicates the normal localization of SCC3 and other meiotic proteins depends on REC8, highlighting REC8 as a pivotal meiotic protein essential for the loading of other meiotic proteins. Additionally, we observed that the end processing of DSBs and the inter-sister/homolog strand invasion processes were profoundly impaired in *rec8*, leading us to speculate that the sticky chromosome phenotype in *rec8* could be attributed to an unknown repair pathway. Our results underscore that SCC3 serves as a cohesin component and plays a significant role in cohesion establishment during both mitosis and meiosis. In contrast, the function of REC8 is only restricted to meiosis. While both proteins operate independently in mitosis, they are interrelated in meiosis, with REC8 emerging as a more critical factor in the meiotic process.

## Materials and methods

### Meiotic chromosome preparation

Young rice panicles were fixed in Carnoy's solution (ethanol: acetic acid, 3: 1, v/v). Microsporocytes at different meiotic stages were squashed in an acetocarmine solution on glass slides. Moreover, the slides were covered with a coverslip and washed with 45% acetic acid from each side. Subsequently, the slides were frozen in liquid nitrogen for 10 s, and then the coverslip was removed as soon as possible ensuring sufficient cells were fixed on the slide. After ethanol gradient dehydration (70%, 90%, and 100%), the slides with chromosome spreads were counterstained with 4',6-diamidino-2-phenylindole (DAPI, Vector Laboratories, Burlingame, CA, USA) solution. Chromosome behaviors were observed and captured using a ZEISS A2 microscope imaging system with a micro-charge-coupled (micro CCD) device camera.

### Antibody production

To generate the antibody against SCC3, a 648 bp C-terminal fragment of *SCC3* complementary DNA (amino acids 910–1116) was amplified using primers SCC3-Ab-F and SCC3-Ab-R (*Supplementary file 1*). The PCR product was cloned into the expression vector pET-30a (Novagen, Madison, WI, USA). The fusion peptide expression and purification were carried out as described previously (*Miao et al., 2021*). The polyclonal antibody was raised from mouse and rabbit. Other antibodies were generated in our laboratory previously.

### Immunofluorescence assay

Fresh young panicles were fixed in 4% (w/v) paraformaldehyde for 30 min at room temperature. Different meiotic stages of anthers were squashed in a drop of 1x PBS solution on glass slides. After freezing in liquid nitrogen and ethanol dehydration, slides were incubated with different combinations of diluted antibodies (1:100) in a humid chamber at 37 °C for 2 hr. Thereafter, the slides were washed with 1x PBS solution three times and were further incubated for 1 hr with the appropriate fluorochrome-coupled secondary antibody, including fluorescein-isothiocyanate-conjugated goat anti-mouse antibody (Southern Biotech, Birmingham, AL, USA), rhodamine-conjugated goat anti-rabbit antibody (Southern Biotech), and AMCA-conjugated goat anti-guinea pig antibody (Jackson Immuno Research, West Grove, PA, USA). Slides were washed with 1x PBS solution three times and eventually stained with DAPI. The immunofluorescence signals were analyzed and captured using a ZEISS A2 microscope imaging system.

### Fluorescence in situ hybridization

Young panicles of both wild-type and mutants were fixated in Carnoy's solution (ethanol: acetic acid, 3: 1, v/v). The FISH analysis was performed as a detailed protocol previously (*Cheng, 2013*). The pAtT4 clone containing telomere repeats, the pTa794 clone containing 5 S ribosomal RNA genes, and the bulked oligonucleotide probes specific to chromosome 12 were used as probes in FISH analysis. Rhodamine anti-digoxigenin was used for digoxigenin-labeled probes. Chromosome images were captured using an Olympus (Shinjuku-ku, Tokyo, Japan) BX51 fluorescence microscope with a micro CCD camera using software IPLAB4.0 (BD Biosciences, San Jose, CA, USA).

### Yeast two-hybrid assay

The yeast two-hybrid (Y2H) assays were conducted by the full-length CDSs of *SCC1*, *SCC3*, *REC8*, *SMC1*, and *SMC3* independently cloned into the pGBKT7 or pGADT7 vector. Co-transformants were selected on SD/DDO (SD-Leu-Trp) medium at 30 °C for 3–4 days. Positive interactions of transformants were selected on SD/QDO (SD-Leu-Trp-His-Ade) medium containing aureobasidin A. All primers used to construct plasmids are listed in *Supplementary file 1*. The detailed protocol was described in the manufacturer's handbook (*Yeast protocols handbook*; PT3024-1; Clontech).

### BiFC assay

To conduct BiFC assays, SCC3 and RAD21.1 were amplified by KOD-plus polymerase and ligated into BiFC vectors, including pSCYNE (SCN) and pSCYCE (SCC). The constructed plasmids were transformed into protoplasts extracted from the young stem of rice-etiolated seedlings. After incubation in

the dark for 18 hr at 28 °C, the CFP signals were captured under a confocal laser scanning microscope at an excitation wavelength of 405 nm (Leica TCS SP5, Wetzlar, Germany).

## STED microscopy and image analysis

Immunofluorescence assays were conducted by the procedures described above. STED images were acquired using Abberior STEDYCON (Abberior Instruments GmbH, Göttingen, Germany) fluorescence microscope built on an upright microscope BX53 (Olympus UPlanXAPO 100 x, NA1.45, Tokyo, Japan). The microscope is equipped with pulsed STED lasers at 775 nm and 640 nm excitation lasers. The appropriate fluorochrome-coupled secondary antibody were used in this study, including abberior STAR ORANGE goat anti-mouse IgG (STORANGE-1001–500 UG), abberior STAR RED goat anti-rabbit IgG (STRED-1002–500 UG). Imaging and image processing was done with ImageJ software.

## RNA extraction and real-time PCR

Real-time PCR assays were used to detect the expression patterns of *SCC3*. Total RNA was extracted from different tissues in wild-type including root, stem, leaf, sheath, spikelet, 2 cm long panicle, 2–3 cm long panicle, 3–5 cm long panicle, 5–7 cm long panicle, and 7 cm long panicle using TRIzol reagent. After digestion with RNase-free DNaseI to remove genomic DNA, reverse transcription PCR was performed to synthesize cDNA using SuperScript III Reverse Transcriptase by manufacturer's protocol (Invitrogen). Quantitative real-time PCR assays were performed on the Bio-Rad CFX96 instrument using Evagreen (Biotium, Freemont, CA, USA) with a procedure of 98 °C 3 min, 40 cycles of 98 °C 15 s, and 60 °C 20 s. The *UBIQUITIN* gene was used as an internal reference. Three biological repeats were performed for each sample. Primers used for real-time PCR assays were listed in *Supplementary file 1*.

## Phylogenetic analysis

The full-length amino acid sequence of SCC3 was selected as a template and searched by the NCBI blastp tool to identify similar proteins among species. A detailed protocol for further analysis has been reported previously (*Li et al., 2018b*; *Ren et al., 2018*; *Ren et al., 2020*). Filtered sequences were downloaded and used for constructing neighbor-joining trees by $M_{EGA}5$ software and processed with EvolvieW (https://evolgenius.info/evolview/).

## Multiple sequence alignments

Multiple alignments were performed using the online toolkit $M_{AFFT}$ (https://toolkit.tuebingen.mpg.de/mafft) and processed with $ESP_{RIPT}3$ (http://espript.ibcp.fr/ESPript/ESPript/) by previously detailed reported (*Zhao et al., 2021*).

## Experimental model and subject details

The rice *scc3* mutant was produced by CRISPR-Cas9 toolkit in *japonica* rice variety Yandao 8. Primers used for CRISPR-Cas9 are listed in *Supplementary file 1*. Other meiosis mutants *pair3* (Wuxiangjing 9), *pair2* (Nipponbare), *zep1* (Nipponbare), *rec8* (Yandao 8), *sgo1* (Yandao 8), *spo11-1* (Guangluai 4), *dmc1* (Nipponbare) have been previously described (*Wang et al., 2016*). Yandao 8 was used as the wild-type. All the rice plants were grown in a paddy field under natural rice-growing conditions.

## Quantification and statistical analysis

At least three independent biological repeats were performed for each experiment. Values in Figures are means ± SD. The significant difference between the wild-type and corresponding mutants was evaluated by two-tailed Student's *t*-test and $p < 0.05$ was considered as a criterion for judging statistically significant differences. The significance labeled in the Figures represents the following meanings: *$p < 0.05$; **$p < 0.01$; ***$p < 0.001$; ****$p < 0.0001$.

## Acknowledgements

This work was supported by grants from the National Key Research and Development Program of China (2023YFA0913500), and grants from the National Natural Science Foundation of China (U2102219, 31930018, and 32201779).

# Additional information

## Funding

| Funder | Grant reference number | Author |
|---|---|---|
| National Key Research and Development Program of China | 2023YFA0913500 | Zhukuan Cheng |
| National Natural Science Foundation of China | U2102219 | Zhukuan Cheng |
| National Natural Science Foundation of China | 31930018 | Zhukuan Cheng |
| National Natural Science Foundation of China | 32201779 | Zhukuan Cheng |

The funders had no role in study design, data collection and interpretation, or the decision to submit the work for publication.

## Author contributions

Yangzi Zhao, Conceptualization, Data curation, Formal analysis, Validation, Writing – original draft, Writing – review and editing; Lijun Ren, Tingting Zhao, Resources, Supervision, Visualization; Hanli You, Yongjie Miao, Lei Cao, Yi Shen, Ding Tang, Supervision, Visualization; Huixin Liu, Supervision, Investigation; Bingxin Wang, Formal analysis, Investigation; Yafei Li, Formal analysis, Supervision, Visualization; Zhukuan Cheng, Conceptualization, Resources, Software, Supervision, Funding acquisition, Validation, Investigation, Visualization, Methodology, Writing – original draft, Writing – review and editing

## Author ORCIDs

Yangzi Zhao ⓘ http://orcid.org/0000-0003-2077-6319
Lijun Ren ⓘ http://orcid.org/0000-0001-5957-1113
Zhukuan Cheng ⓘ https://orcid.org/0000-0001-8428-8010

Reviewer #1 (Public Review): https://doi.org/10.7554/eLife.94180.3.sa1
Reviewer #2 (Public Review): https://doi.org/10.7554/eLife.94180.3.sa2
Reviewer #3 (Public Review): https://doi.org/10.7554/eLife.94180.3.sa3
Author response https://doi.org/10.7554/eLife.94180.3.sa4

# Additional files

## Supplementary files

• Supplementary file 1. List of primers used in this study.

• MDAR checklist

## Data availability

All raw microscopy imagine data generated or analysed during this study are uploaded to Dryad at https://doi.org/10.5061/dryad.kh18932fz.

The following dataset was generated:

| Author(s) | Year | Dataset title | Dataset URL | Database and Identifier |
|---|---|---|---|---|
| Cheng Z | 2024 | SCC3 is an axial element essential for homologous chromosome pairing and synapsis | https://doi.org/10.5061/dryad.kh18932fz | Dryad Digital Repository, 10.5061/dryad.kh18932fz |

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
