## [Editor Report · eLife assessment]

This **fundamental** study elucidates the function of the cohesin subunit SCC3 in maintaining homologous chromosome pairing and synapsis during meiosis. The observation of sterility in the SCC3 weak mutant prompted an investigation of abnormal chromosome behavior during anaphase I, and the discovery that SCC3's loading onto meiotic chromosomes is REC8-dependent. The **convincing** evidence presented in this study contributes to our understanding of meiosis in rice and attracts cell biologists, reproductive biologists, and plant geneticists.

---

## [Referee Report · Reviewer #1 (Public Review)]

Summary:

The revised manuscript is much improved. As stated previously, it is on an interesting and important topic and provides many new potentially important findings. The manuscript contains a large amount of high-quality data. In the revised manuscript, the authors have done a nice job addressing the concerns raised in the previous review. They have refined their conclusions and the evidence provided supports conclusions drawn. Likewise, the writing and low of the manuscript is much improved.

Strengths:

The manuscript contains a large amount of high-quality data that is used to draw interesting and important conclusions.

Weaknesses:

There are still some issues with grammar and word usage, but these should be easily corrected with some additional minor editing. Other than some minor editing, my only real question/concern is whether the data presented shows that SCC3 is directly involved in gene regulation. It may well be that changes in chromatin structure caused by mutations in SCC3 and the axial element protein containing genes examined indirectly affect transcript levels for the genes examined.

---

## [Referee Report · Reviewer #2 (Public Review)]

Summary:

This manuscript shows detailed evidence about the role of cohesin regulator in rice meiosis and mitosis

Strengths:

There is a very clear mechanism for its role during replication

Weaknesses:

The authors did not consider to create heterozygous mutants for the replication fork.

April 15. Revisions read.

---

## [Referee Report · Reviewer #3 (Public Review)]

Summary:

Prior research on SCC3, a cohesin subunit protein, in yeast and Arabidopsis has underscored its vital role in cell division. This study investigated into the specific functions of SCC3 in rice mitosis and meiosis. In a weakened SCC3 mutant, sister chromatids separating was observed in anaphase I, resulting in 24 univalents and subsequent sterility. The authors meticulously documented SCC3's loading and degradation dynamics on chromosomes, noting its impact on DNA replication. Despite the loss of homologous chromosome pairing and synapsis in the mutant, chromosomes retained double-strand breaks without fragmenting. Consequently, the authors inferred that in the scc3 mutant, DNA repair more frequently relies on sister chromatids as templates compared to the wild type.

Strengths:

The study presents exceptionally well-executed research in the field of rice cytogenetics.

Weaknesses:

While the paper's conclusions are generally well-supported, further substantiation is needed for the claim that SCC3 inhibits template choice for sister chromatids. To bolster this conclusion, I recommend that the authors perform whole-genome sequencing on parental and F1 individuals from two rice variants, subsequently calculating the allele frequencies at heterozygous sites in the F1 individuals. If SCC3 indeed inhibits inter-sister chromatid repair in the wild type, we would anticipate a higher frequency of inter-homologous chromosome repair (i.e., gene conversion). This should be manifested as a bias away from the Mendelian inheritance ratio (50:50) in the offspring of the wild type compared to the offspring of the scc3+/- mutant.

---

## [Author Response]

The following is the authors’ response to the original reviews.

We express our sincere appreciation for your insightful comments and constructive suggestions. It is with great pleasure that we submit the revised version of our manuscript. Over the past months, we have meticulously considered all the invaluable feedback provided by the three anonymous reviewers, and endeavored to incorporate significant revisions accordingly. Furthermore, we have meticulously rephrased the results section in accordance with your guidance, aiming to bolster the rigor of our manuscript. The specific changes implemented in the revised manuscript are outlined below:

- Revised the title of the manuscript.

- Revised the description of early mitotic and meiotic chromosome structure in the *scc3* mutant (Lines 167-274).

- Added the BiFC results illustrating the interaction between SCC3 and other cohesin proteins in Figure S10.

- Enhanced the detail in the description of figure legends, particularly for Figures 2 and 4.

- Refined and rephrased the language of the manuscript.

We hope these positive revisions have substantially strengthened the manuscript. Once again, we extend our heartfelt gratitude for your invaluable input.

**eLife assessment**
This important study elucidates the function of the cohesin subunit SCC3 in impeding DNA repair between inter-sister chromatids in rice. The observation of sterility in the SCC3 weak mutant prompted an investigation of abnormal chromosome behavior during anaphase I through karyotype analysis. While the evidence presented is largely solid, the strength of support can be substantially improved in some aspects, leaving room for further investigation. This research contributes to our understanding of meiosis in rice and attracts cell biologists, reproductive biologists, and plant geneticists.
**Public Reviews:**

**Reviewer #1 (Public Review):**
The manuscript describes the identification and characterization of rice SCC3, including the generation and characterization of plants containing apparently lethal null mutations in SCC3 as well as mutant plants containing a c-terminal frame-shift mutation. The weak scc3 mutants showed both vegetative and reproductive defects. Specifically, mitotic chromosomes appeared to partially separate during prometaphase, while meiotic chromosomes were diffuse during early meiosis and showed alterations in sister chromatid cohesion, homologous chromosome pairing, and recombination. The authors suggest that SCC3 acts as a cohesin subunit in mitosis and meiosis, but also plays more functions other than just cohesion.
**Reviewer #2 (Public Review):**
This manuscript shows detailed evidence of the role of cohesin regulators in rice meiosis and mitosis.
**Reviewer #3 (Public Review):**
Prior research on SCC3, a cohesin subunit protein, in yeast and Arabidopsis has underscored its vital role in cell division. This study investigated into the specific functions of SCC3 in rice mitosis and meiosis. In a weakened SCC3 mutant, sister chromatids separating was observed in anaphase I, resulting in 24 univalents and subsequent sterility. The authors meticulously documented SCC3's loading and degradation dynamics on chromosomes, noting its impact on DNA replication. Despite the loss of homologous chromosome pairing and synapsis in the mutant, chromosomes retained double-strand breaks without fragmenting. Consequently, the authors inferred that in the scc3 mutant, DNA repair more frequently relies on sister chromatids as templates compared to the wild type.

We extend our sincere gratitude to the Editors and the Reviewers for their highly constructive and insightful suggestions. We deeply appreciate receiving both positive feedback and constructive criticism on our manuscript. In light of the reviewers’ comments, we have diligently undertaken substantial revisions to improve the manuscript. The revised version comprehensively addresses all the points raised by the reviewers.

Below, we provide a detailed point-by-point response to the reviewers’ comments:

**Recommendations for the authors:**

**Reviewer #1:**
(1) Line 170- looking at pollen formation does not specifically evaluate whether SCC3 is involved in meiosis.

Thank you very much for this advice. We totally agree with your point of view that pollen formation defects only indicate the problem of gametogenesis. We are sorry for not accurately describing this sentence. It has been revised in the manuscript (Lines 167-176).

(2) Lines 203-205- this seems more like discussion and is pure speculation. Another possibility described above is that the truncated SCC3 protein is partially functional and what they see is due to this partial functionality. Have the authors considered the possibility that a partially functional version of SCC3 is produced that alters its function or the function of the cohesin complex? How much of the protein epitope remains in the truncated protein?

We are so grateful for the insightful suggestions provided. We concur with the proposition that a partially functional SCC3 may indeed be synthesized, contributing to the survivability of the mutant. Notably, the truncated version of the protein retains approximately 60% to 70% of the epitope, which ostensibly maintains a residual functionality within the weak *scc3* mutant. In this manuscript, the loss of C-terminal 910-1116 aa of SCC3 contains a special protein epitope and a certain protein secondary structure, which may alter the protein’s folding and its subsequent roles within the cohesin complex.

In this study, we encountered challenges in generating null alleles of the *scc3* mutants in rice utilizing the CRISPR-Cas9 system. Consequently, it is plausible that the *scc3-1* and *scc3-2* variants represent null alleles of SCC3, resulting in embryonic lethality. We posit that the identification of weak alleles is paramount to facilitating the survival of the organism. Thus, selecting some weak mutants, particularly those exhibiting the most pronounced phenotype, is advantageous for conducting further research. Our findings indicate that the diminished *scc3* mutant lacks only a segment of the C-terminal, yet this deficiency is adequate to ensure the plant's survival while significantly impeding the meiotic process. We cannot dismiss the likelihood that these observed defects are attributable to the unique truncated proteins. We extend our sincerest thanks once again.

(3) Lines 212- I question whether what the authors see in Figure 2 is chromosome fragmentation. It could just as well be alterations in chromosome structure. Likewise, the authors provide little to no evidence that the mutation affects the replication process. Rather, the presence of replicated chromosomes later in mitosis and meiosis would argue that replication is not disrupted.

We express our gratitude to the reviewer for highlighting this critical inquiry. Contrary to the scenario of chromosome fragmentation, as you astutely observed, the preservation of normal sister chromatids during prometaphase indicates that the replication process remains uninterrupted. In alignment with your insights, our study embarked on an extensive series of full-length fluorescence in situ hybridization (FISH) experiments to elucidate the underlying mechanisms contributing to the observed increase in the distance between sister chromatids, particularly during interphase. The preponderance of our findings corroborates the hypothesis that the chromosomes exhibit alterations in structure, as depicted in Figure 2A. Intriguingly, our data suggest that cohesin, upon interaction with other chromatin-bound proteins, may facilitate loop extrusion, anchoring itself in a manner that potentially alters chromosomal architecture. These alterations in chromosome structure and the subsequent defects in genome folding and cohesion establishment, particularly rely on SCC3. In response to your valuable suggestions, we have meticulously revised the relevant sections of our manuscript. We extend our sincere thanks for your insightful comments.

(4) Line 230- what does the sentence SCC3 may enhance the interaction with DNA mean, the interaction of the cohesin complex?

We are sorry for the ambiguity in our initial description and wish to clarify that SCC3 indeed plays a pivotal role in augmenting the interaction between the cohesin complex and DNA. Our observations revealed an upsurge in the signal intensity of SCC3 as cells transition from interphase to prophase, as depicted in Figure 2B. This enhancement correlates with the observed defects in *scc3* mutants during prophase, suggesting that SCC3’s functional significance is particularly pronounced at this stage of the cell cycle. We have revised our manuscript to reflect these insights more accurately, in accordance with your valuable suggestions. We express our sincere gratitude for your guidance.

(5) Oddly, and unexplainably the authors present data indicating that SCC3 interacts with RAD21.1, but not SMC1, SMC3, or REC8. The fact that the authors report that SCC3 only interacts with RAD21.1 but no other cohesin proteins is quite hard to explain.

As argued in the point above, the available data do not provide compelling evidence supporting the interaction between SCC3 and other cohesin proteins. We have repeated yeast two-hybrid (Y2H) experiments yielding consistent outcomes, which also surprised us initially. In the revised manuscript, we further added the bimolecular fluorescence complementation (BiFC) results between SCC3 and other cohesin proteins in rice protoplast (Figure S10). These supplementary data affirm that SCC3 predominantly interacts with RAD21.1, excluding interactions with other cohesin proteins. While the absence of such interactions is perplexing, our investigations have failed to detect any binding between SCC3 and other cohesin proteins.

A weak interaction between SCC3 and REC8 has been reported in Arabidopsis (Kuttig et al. bioRxiv https://doi.org/10.1101/2022.06.20.496767). We speculate that either these proteins do not interact or the yeast-hybrid assays may be inadequate for detecting their interaction, as several factors can impede interaction in a heterologous system. In Figure 7, we could only detect the interaction between SCC3 and RAD21.1 in both Y2H and BiFC experiments. This suggests potential alterations in protein folding or conformation, or the involvement of additional regulatory factors modulating the interaction between SCC3 and other cohesin proteins. Notably, given RAD21.1’s pivotal role as a core component in the cohesin complex, our supplementary findings demonstrate the interactions between SMC1/3 and RAD21.1 (data not shown). Consequently, our current data propose a model wherein RAD21.1 and SMC1/3 form a circular structure, with SCC3 positioned on the outer periphery of the ring complex, associating specifically with RAD21.1 (Figure 8A).

**Reviewer #2:**
The authors did not consider creating heterozygous mutants for the replication fork. Moderate English language editing may be required.

We extend our gratitude to the reviewer for their valuable suggestions. Initially, we did not explore the potential relationship between SCC3 and the replication fork. Cohesin, as we understand, becomes associated with DNA prior to DNA replication. The phenomenon of sister chromatid co-entrapment arises as replication forks traverse through cohesin rings, a process intricately linked to DNA replication dynamics. In this study, we exclusively observed aberrant chromosome structures in the *scc3* mutant during interphase (Figure 2). We conjecture that these anomalies may stem from alterations in chromosome structure, such as genome folding and loop extrusion, rather than being directly attributable to the DNA replication fork. However, the precise nature of these chromosome structural aberrations during interphase in the *scc3* mutant remains elusive, necessitating further comprehensive investigation in future studies. We have refined the language of our manuscript in accordance with the reviewer’s suggestions. Once again, we express our sincere appreciation for the invaluable suggestions provided.

**Reviewer #3:**
While the paper's conclusions are generally well-supported, further substantiation is needed for the claim that SCC3 inhibits template choice for sister chromatids. To bolster this conclusion, I recommend that the authors perform whole-genome sequencing on parental and F1 individuals from two rice variants, subsequently calculating the allele frequencies at heterozygous sites in the F1 individuals. If SCC3 indeed inhibits inter-sister chromatid repair in the wild type, we would anticipate a higher frequency of inter-homologous chromosome repair (i.e., gene conversion). This should be manifested as a bias away from the Mendelian inheritance ratio (50:50) in the offspring of the wild type compared to the offspring of the scc3+/- mutant.

We express our sincere appreciation for your insightful suggestions. It is really a good suggestion. We have arranged to do this experiment. As it takes long time to prepare plant materials and sequence analysis, we hope the ongoing sequencing work will get some important information supporting those hypotheses. As we have not obtained the direct evidence that SCC3 involved in sister chromatid repair, we changed the title as “SCC3 is an axial element essential for homologous chromosome pairing and synapsis”. Once again, we really extend our gratitude for your invaluable suggestions.

A point that warrants consideration is the placement of the protein interaction experiments involving SCC3 within the paper. It is presented relatively late in the manuscript. If the authors possess information regarding the interaction between RAD21.1 and SCC3 and how it relates to the functional study of RAD21.1, it could contribute to a more comprehensive analysis. However, if this information is unrelated to the current study, it might be advisable to omit it, as it appears to diverge from the main focus of this work.

We express our sincere gratitude for your invaluable suggestions. It has been documented in yeast that the interaction between SCC3 and SCC1 is indispensable for the efficient loading of cohesin. In our study, we endeavored to elucidate the intricate relationships among various cohesin subunits. Through our investigations, we have discerned that RAD21.1 serves as a pivotal core subunit within the cohesin complex, facilitating interactions with both SMC1/3 and SCC3 (data not shown). Additionally, our findings indicate that the interaction between RAD21.1 and SCC3 is imperative for maintaining the stability of the cohesin ring and its association with DNA (data not shown). Consequently, the interaction between these two proteins assumes paramount importance for our subsequent analyses. This study holds significant promise for future investigations.

It's worth noting that while the title of the study claims that "SCC3 inhibits inter-sister chromatids repair during rice meiosis," the last sentence of the abstract weakens this conclusion by using the word "seems." A study's title should ideally reflect the most definitive and conclusive findings.

We sincerely appreciate your valuable suggestions. In response, we have revised the description in our manuscript to enhance its rigor.

In Figure 8C, it appears that cohesin is depicted between two DNA strands.

Figure 8C illustrates the process of sister chromatid repair during meiosis in the *scc3* mutant. Two gray lines and two blue lines represent the four sister chromatids of two homologous chromosomes, respectively. In the wild type, cohesin plays a crucial role in tethering together the two sister chromatids. As per your reminder, cohesin should indeed encircle the two sister chromatids, as depicted in Figure 8B. Following a thorough evaluation and to mitigate any potential confusion, we have deleted Figure 8C.